# Improving Hepatocellular Carcinoma Surveillance Outcomes in Patients with Cirrhosis after Hepatitis C Cure: A Modelling Study

**DOI:** 10.3390/cancers16152745

**Published:** 2024-08-01

**Authors:** Jacob Cumming, Nick Scott, Jessica Howell, Joan Ericka Flores, Damian Pavlyshyn, Margaret E. Hellard, Leon Shin-han Winata, Marno Ryan, Tom Sutherland, Alexander J. Thompson, Joseph S. Doyle, Rachel Sacks-Davis

**Affiliations:** 1Disease Elimination Program, Burnet Institute, Melbourne, VIC 3004, Australiajoseph.doyle@burnet.edu.au (J.S.D.); rachel.sacks-davis@burnet.edu.au (R.S.-D.); 2Population Health and Immunity, Walter and Eliza Hall Institute, Parkville, VIC 3052, Australia; 3Department of Epidemiology and Preventive Medicine, Monash University, Melbourne, VIC 3004, Australia; 4Department of Gastroenterology, St Vincent’s Hospital, Melbourne, VIC 3065, Australia; 5Department of Medicine, University of Melbourne, Melbourne, VIC 3052, Australia; 6Department of Infectious Diseases, The Alfred and Monash University, Melbourne, VIC 3004, Australia; 7Doherty Institute and School of Population and Global Health, University of Melbourne, Melbourne, VIC 3052, Australia

**Keywords:** hepatocellular carcinoma, epidemiological models, liver cirrhosis, Hepatitis C, disease progression, epidemiologic surveillance, early detection of cancer, life expectancy, alpha-fetoproteins

## Abstract

**Simple Summary:**

The mortality of hepatocellular carcinoma (HCC) is rising globally, against the trend of other cancers. People with liver cirrhosis, even after hepatitis C treatment still face a high risk of HCC, requiring ongoing enrolment in HCC surveillance, and new technologies to improve diagnostic sensitivity are being explored. However, their impact on HCC survival remains uncertain relative to improving adherence to existing surveillance methods. This study uses mathematical modeling to assess how different strategies can reduce deaths from liver cancer in people with cirrhosis after being cured of hepatitis C. We compared the impact of improved adherence to ultrasound screening with increased HCC imaging sensitivity on HCC survival. Notably, we found that even modest enhancements in surveillance adherence (5–10 percentage point increases) exhibited significant survival benefits for people with hepatitis C-related cirrhosis, outperforming improvements in diagnostic sensitivity.

**Abstract:**

Background & Aims: Hepatocellular carcinoma (HCC) presents a significant global health challenge, particularly among individuals with liver cirrhosis, with hepatitis C (HCV) a major cause. In people with HCV-related cirrhosis, an increased risk of HCC remains after cure. HCC surveillance with six monthly ultrasounds has been shown to improve survival. However, adherence to biannual screening is currently suboptimal. This study aimed to evaluate the effect of increased HCC surveillance uptake and improved ultrasound sensitivity on mortality among people with HCV-related cirrhosis post HCV cure. Methods: This study utilized mathematical modelling to assess HCC progression, surveillance, diagnosis, and treatment among individuals with cirrhosis who had successfully been treated for HCV. The deterministic compartmental model incorporated Barcelona Clinic Liver Cancer (BCLC) stages to simulate disease progression and diagnosis probabilities in 100 people with cirrhosis who had successfully been treated for hepatitis C over 10 years. Four interventions were modelled to assess their potential for improving life expectancy: realistic improvements to surveillance adherence, optimistic improvements to surveillance adherence, diagnosis sensitivity enhancements, and improved treatment efficacy Results: Realistic adherence improvements resulted in 9.8 (95% CI 7.9, 11.6) life years gained per cohort of 100 over a 10-year intervention period; 17.2 (13.9, 20.3) life years were achieved in optimistic adherence improvements. Diagnosis sensitivity improvements led to a 7.0 (3.6, 13.8) year gain in life years, and treatment improvements improved life years by 9.0 (7.5, 10.3) years. Conclusions: Regular HCC ultrasound surveillance remains crucial to reduce mortality among people with cured hepatitis C and cirrhosis. Our study highlights that even minor enhancements to adherence to ultrasound surveillance can significantly boost life expectancy across populations more effectively than strategies that increase surveillance sensitivity or treatment efficacy.

## 1. Introduction

Globally, hepatocellular carcinoma (HCC) accounts for approximately 90% of all primary liver cancers and is the leading cause of death for those with liver cirrhosis [1]. In 2021, liver cancer in Australia was diagnosed in over 2800 people, was responsible for more than 2400 deaths, and had one of the lowest five-year survivals for any cancer type [2]. Hepatitis C is a lead cause of HCC globally [3], and until recently was the lead cause in Australia [2]. By 2019, hepatitis C also accounted for around one in four liver transplantations in Australia and New Zealand [4]. Improving surveillance and prevention of HCC among people with hepatitis C and after cure remains an ongoing clinical and population health priority/challenge.

Around 1% of those with hepatitis C will progress to cirrhosis, and without a hepatitis C cure, approximately 5% of that group will progress to HCC per year [5]. Since direct-acting antiviral (DAA) treatments have become available, as of the end of 2020, Australia had treated approximately 60% of people with hepatitis C, with approximately 75,000 people living with the virus [6]. This substantially reduces their risk of developing HCC. However, for those with cirrhosis, a residual risk (around 2.2% per annum) of developing HCC remains even after achieving a sustained virologic response (SVR) [1,7].

There is good evidence that HCC surveillance with six monthly ultrasounds, with or without alpha-fetoprotein measurement, increases the probability of earlier-stage diagnosis of HCC, thereby improving survival [8]. Therefore, this is recommended in clinical guidelines for people with hepatitis C-related cirrhosis after achieving sustained virologic response [7,9,10,11].

Few studies have explored how interventions improving HCC surveillance impact HCC survival. While the cost-effectiveness of HCC surveillance with biannual ultrasound is well established [12,13,14], the relative impact of strategies to increase HCC surveillance sensitivity, increase current HCC surveillance uptake, or increase treatment effectiveness on HCC survival is unknown. Furthermore, many modelling studies of the impact of HCC surveillance on survival have assumed perfect adherence to surveillance [12,13,14,15]. In practice, imperfect adherence and potentially low ultrasound sensitivity, as seen in real-world settings, are likely to substantially reduce the effectiveness of HCC surveillance, particularly with the increasing prevalence of MASLD and obesity, which can impact ultrasound sensitivity for HCC [16]. New methods of HCC surveillance designed to improve sensitivity for early HCC detection, such as rapid sequence non-contrast MRI and the addition of blood-based biomarkers such as GALAD score, are being actively explored [16]. Still, their differential impact on HCC survival has not been evaluated.

### Aims

In this study, we aimed to estimate the impact of two potential public health interventions on HCC-related mortality among people with cirrhosis who were successfully treated for hepatitis C: increasing HCC surveillance uptake and improved ultrasound sensitivity. We also considered modest improvements in HCC clinical treatment efficacy as an alternative intervention for a relative comparison.

## 2. Methods

### 2.1. Model Structure

This is a mathematical modelling study of HCC progression, surveillance, diagnosis, and treatment among people with cirrhosis. Our model measures the number of life years extended by treatment for 100 simulated people with cirrhosis over 10 years (unless otherwise specified). It is a deterministic compartmental model that accounts for the sensitivity of ultrasound detection of HCC at each Barcelona Clinic Liver Cancer (BCLC) stage (Table 1). The model has monthly discrete time steps. Model compartments represent the BCLC stages of cancer (with the two additional compartments: cirrhosis without HCC and death). Each time step, a BCLC stage-dependent proportion of undiagnosed people in each compartment transition to the next BCLC stage (Figure 1).

Depending on the BCLC stage of cancer, the model has a background probability of a HCC diagnosis either because of symptoms or incidentally through unrelated medical scans (such as magnetic resonance imaging (MRI)).

In addition, the model includes scheduled surveillance ultrasound attendance. Simulated people are distributed into one of three different cohorts (according to proportions from model parameters) representing their adherence to surveillance guidelines: a completely adherent, partially adherent, and non-adherent cohort. Those in the completely adherent and non-adherent cohorts always and never attended scheduled surveillance. Those in the partially adherent cohort attended ultrasound visits with a specified probability. For the proportion of those who attended their scheduled ultrasounds, sensitivity depended on the individual’s BCLC stage. If HCC was detected, that individual was marked as diagnosed and can no longer progress to further stages. Ultrasound sensitivities were assumed to be consistent across the cohorts, irrespective of the level of adherence.

At the end of the 10-year simulation, the number of people diagnosed in each BCLC stage was multiplied by an estimated additional life expectancy gained through diagnosis (dependent on the BCLC stage) to estimate the total number of life years gained. We assumed that any additional life expectancy gained through diagnosis was due to treatment. This value was calculated by comparing the life expectancy of treated and untreated individuals by BCLC stage [17]. Table 2 reports these values.

### 2.2. Data and Calibrations

#### 2.2.1. Monthly Transition Probabilities, Ultrasound Sensitivity, Symptom Likelihood

We derived BCLC stage-dependent monthly transition probabilities, ultrasound sensitivities, and the likelihood of developing symptoms using data from the literature and by numerically solving for the maximum likelihood estimate of the corresponding model parameters [8]. We estimated the uncertainty in the model parameters by creating 10,000 bootstrap resamples of the data and finding the maximum likelihood estimate for each resample. This is described in more detail in the Appendix A, including Appendix A, and Appendix A.

#### 2.2.2. Data Sources

To our knowledge, there are no published estimates of the monthly rate at which people transition between the BCLC stages of HCC, so these results were calibrated from median time until death, stratified by the BCLC stage described by Giannini and colleagues [25].

Ultrasound sensitivity has previously been described for groups of BCLC stages (0/A vs. B/C/D) [23,24,26,27]. We used the observed distribution of BCLC stage at diagnosis in a cohort of Canadians diagnosed with HCC who were adherent to timely surveillance (n=109) and the monthly transition probabilities to calibrate BCLC stage-specific ultrasound sensitivity for each BCLC stage.

The monthly probability of developing symptoms was estimated from an Australian multi-site study, as reported by Hong and colleagues [8]. Of those not in surveillance, 33% (52 of n=156) diagnosed with HCC were diagnosed in BCLC stage A/B (as opposed to C/D). This was taken as a proxy for diagnosis of HCC due to symptoms (or some other incidental diagnosis). We also assumed that ultrasound sensitivity for BCLC stage B was at least as good as that at BCLC stage A. Since the model to reproduce these data is non-identifiable (because the data do not distinguish between symptoms at BCLC stages A and B), the likelihood is not maximized at a unique point. Therefore, we chose the point estimate to be the maximum likelihood estimate closest to the center of the two maximum likelihood estimates that were the most extreme. For more details, see the Appendix A.

To calibrate the adherence model parameters, we used unpublished surveillance data from a tertiary hospital in Australia. The methods and results for these data are in the Appendix A, and Appendix A.

### 2.3. Relative Impact of Changes to Modifiable Factors on Life Years Gained

To inform the development of intervention scenarios, the following were identified as factors that could plausibly be targeted for intervention: (1) adherence to ultrasound surveillance recommendations, (2) ultrasound sensitivity for BCLC stages 0 to C HCC, and (3) clinical treatments for BCLC stages 0 to C HCC. To identify which would be most impactful, each modifiable variable was varied one at a time by ±25% relative to the point estimate, and the change in average life years saved from treatment was calculated.

### 2.4. Development of Specific Intervention Scenarios

Four potential intervention scenarios were developed for analysis. These were informed by preliminary modelling results and clinical expertise (of specialists in gastroenterology, public health, and infectious diseases) around which interventions were likely most practical to implement. Note that all percentage point increases are absolute (as opposed to the relative changes in the previous section).

Scenario (1a), which we called the *realistic adherence improvements*, consisted of a five-percentage point decrease in non-adherence, a ten-percentage point increase in complete adherence, and a ten-percentage point increase in the probability that a person in the partially adherent cohort attends a scheduled ultrasound. Previous studies have shown that improvements of 5–15 percentage points in surveillance uptake are reasonable to achieve at a population level [28]. We have assumed, as is the case in Australia and similar high-income countries with universal healthcare, that ultrasound surveillance scans are of no direct cost barrier to the patient due to government funding.

Scenario (1b), which we called *optimistic adherence improvements*, consisted of a 10-percentage point decrease in non-adherence, a 15-percentage point increase in complete adherence, and a 20-percentage point improvement in the probability that a person in the partially adherent cohort attends a scheduled ultrasound.

Scenario (2), which we called *diagnosis sensitivity improvements*, consisted of a five-percentage-point increase in ultrasound sensitivity for BCLC stages 0, A, and B.

Scenario (3), which we called *treatment improvements*, consisted of a one-year increase in life expectancy for those treated with BCLC stage A HCC and a six-month increase for those treated with BCLC stage B. New systemic therapies and adjuvant therapy combinations are being evaluated. Therefore, we included the potential impact of future treatment efficacy improvements in our model.

For each scenario, the number of life years gained per cohort of 100 over a 10-year intervention period was quantified compared to the status quo.

### 2.5. Sensitivity and Uncertainty Analyses

Although the rate of progression from cirrhosis to BCLC stage 0 HCC is fixed in the model, in reality, the rate of liver disease progression varies by co-morbidity status. To account for this, we tested the impact of varying the rate of progression from cirrhosis to HCC on intervention effectiveness. We compared annual risks of 1.1% (low), 2.2% (high), and 2.2% (baseline).

Although the rate of progression from cirrhosis to BCLC stage 0 HCC is fixed in the model, there are many co-morbidities. To account for this, we ran the model with differing rates of progression to HCC to evaluate the impact on intervention effectiveness.

We estimated the model’s uncertainty due to the model parameters’ estimation by calculating the standard error of the number of life years extended by treatment (for 100 simulated people with cirrhosis over ten years) using the parametric bootstrap estimates. We identified which of the model parameters contributed most significantly to the variance in outcome by comparing the partial correlation coefficients of the model parameters $p_{0}$, $p_{A}$, $p_{B}$, $p_{C}$, $p_{D}$, ${Sens}_{0}$, ${Sens}_{A}$, and ${Sym}_{A}$ after logit transformation with respect to the model without intervention. The partial correlation between two random variables is the correlation between these variables after correcting for the other (logit transformed) variables which may be highly correlated with the parameter of interest.

## 3. Results

### 3.1. Data and Calibrations

#### Monthly Transition Probabilities, Ultrasound Sensitivity, Symptom Likelihood

The calibrated distributions for the mean years in each BCLC stage are presented in Figure 2A. Most time was spent in BCLC stages 0 and A, with the mean time spent in these stages of 12.5 (95% CI 4.2, 32.6) and 18.0 (12.2, 23.5) months, respectively. BCLC stage B, C, and D had an average duration of 2.9 (1.4, 4.9), 2.8 (1.7, 4.5), and 6.2 (6.1, 7.8) months. The average time spent in stage A was greater than the average time spent in stages B, C, and D for all bootstrap estimates and was greater than the average time spent in stage 0 in 73% of bootstrap estimates.

For point estimates and 95% confidence intervals for the monthly probability of progressing from each BCLC stage to the next ($p_{0}$, $p_{A}$, $p_{B}$, $p_{C}$, and $p_{D}$), ultrasound sensitivity for BCLC stages 0 and A (${Sens}_{0}$ and ${Sens}_{A}$), and monthly probability of developing symptoms at stages A and B (${Sym}_{A}$ and ${Sym}_{B}$), see Table 3 and Figure 2B. For further details, see the Appendix A.

### 3.2. Relative Impact of Changes to Modifiable Factors on Life Years Gained

Improving ultrasound adherence gained 3.5, 4.4, and 3.6 life years per cohort of 100 persons with cirrhosis over 10 years for attendance probability within partial adherents, decrease in non-adherence, and increase in complete adherence, respectively.

Improvements in ultrasound sensitivity were most significant at early BCLC stages, with 6.6 (BCLC stage 0) and 1.7 (BCLC stage A) life years gained per cohort of 100 persons with cirrhosis over 10 years. In contrast, only 0.2 and 0.005 life years were added due to improvements in ultrasound sensitivity at BCLC stages B and C, respectively, per cohort of 100 persons with cirrhosis over a period of 10 years.

Improvements to treatment quality were most significant for treatments in BCLC stages 0 and A (Figure 2D). Improving treatment quality for BCLC stage 0 and A diagnoses resulted in 10.2 and 5.7 life years gained per cohort of 100 persons over a 10-year intervention period, respectively. In contrast, improvements to treatment quality at BCLC stages B and C only resulted in 0.95 and 0.2 life years gained per cohort of 100 persons with cirrhosis over a 10-year period.

### 3.3. Life Years Gained by Specific Intervention Scenarios

Improving ultrasound adherence was more effective than improvements in diagnostic sensitivity or HCC treatment effectiveness (Table 4 and Figure 3). The *realistic adherence improvements* resulted in 9.8 (95% CI 7.9, 11.6) life years gained per cohort of 100 over a 10-year intervention period; 17.2 (13.9, 20.3) life years were added in *optimistic adherence improvements*. *Diagnosis sensitivity improvements* led to a 7 (3.6, 13.8) year gain in life years, and *treatment improvements* improved life years by 9 (7.5, 10.3) years. Life years gained by *realistic adherence improvements* were greater than *diagnosis sensitivity improvements* and *treatment improvements* in 80% and 74% of simulations (using the bootstrap estimates for the model parameters).

### 3.4. Sensitivity and Uncertainty Analysis

The rate of progression from cirrhosis to HCC impacts the absolute effectiveness of each intervention but does not change the relative effectiveness. In other words, an increased risk of developing HCC leads to a proportionately equal increase in the effectiveness of all interventions, and vice versa for a decreased risk. For all cirrhosis to HCC progression rates, the relative efficacy of the two adherence interventions was greater than improvements in diagnostic sensitivity or treatment effectiveness. See the Appendix A for more details, including Appendix A.

The number of life years extended by treatment (for 100 simulated people with cirrhosis over a 10-year period) had a standard error of 5.7 years when calculated using the bootstrap estimates (see Appendix A for more detail). The most influential model parameters were the monthly probability of progression from BCLC stage 0 to A ($p_{0}$), which had a partial correlation coefficient of −0.87 with life years gained (the monthly probability of transition from BCLC stage 0 to A was negatively correlated with life years gained after adjusting for confounding effects from other variables), and the ultrasound sensitivity at stage 0 (${Sens}_{0}$), which had a partial correlation coefficient of 0.92 with life years gained (a strong positive correlation between ultrasound sensitivity in BCLC stage 0 and life years gained). For more parameters, see Figure 2C.

## 4. Discussion

Our model highlighted the importance of strategies to increase regular participation in surveillance programs and produced quantitative estimates of their impact on life expectancy. Importantly, we showed that even modest improvements in surveillance adherence could improve life expectancy at a population level. We demonstrated that moderate improvements to surveillance adherence were more effective than improvements to treatment efficacy or diagnostic accuracy, which was robust to heterogeneity in HCC risk. Overall adherence of 47% (unpublished data) is comparable to but slightly lower than Australian participation rates for breast cancer surveillance (around 55% in 2018–19 with surveillance once every two years for women aged 50–74) and cervical surveillance (around 60% in 2018–2021 with surveillance recommended once every five years for women between 25 and 74) [29,30]. These findings are significant because policy measures to improve HCC surveillance uptake could realistically be introduced for this targeted population, analogous to other national strategies that have achieved earlier (curative stage) cancer detection in much larger population cohorts.

We have shown that improvements in adherence to surveillance have the potential to significantly improve the life expectancy of people with cirrhosis who have been cured of hepatitis C. In our model, this was due to increases in the frequency of ultrasounds, which led to a higher probability of earlier detection and, therefore, a better prognosis. In addition, a meta-analysis by Singal et al. suggested that regular surveillance could additionally lead to increased ultrasound sensitivity [27]. Conservatively, we did not account for this possible effect in the model. Adherence has generally been shown to be higher for those who are screened in subspecialty gastroenterology/hepatology clinics [31,32]. Considering these factors, policymakers should consider the possibility of more significant investment in existing subspecialty liver clinic surveillance programs to enhance retention and timely adherence, or coordination of HCC surveillance programs via specialist services (such as via a hub and spoke model), and a high priority should be given towards patient retention. This may require changes to systems that track those who need regular ultrasounds, including regular follow-up for those who miss scheduled ultrasounds, mainly targeting the significant drop off in attendance after the first year of scheduled surveillance [33].

Kennedy and colleagues have demonstrated cost-effective strategies, including ‘improved doctor education, system redesign, and improved patient education,’ that improved adherence in a tertiary care hospital in Adelaide, Australia [34]. More work is needed to show the cost-effectiveness of modest interventions, which are likely to be dependent on the setting, and the cost of the particular intervention chosen. This work should include understanding whether local strategies are scalable and cost-effective at the system level. Current surveillance adherence is suboptimal in Australia [8,35]. The surveillance uptake at the hospital site used to inform the model was consistent with surveillance adherence for other Australian and European tertiary hospitals, as reported elsewhere in the literature, indicating the scope for improvement at many sites [36,37].

Our model results suggest that improvements in ultrasound sensitivity and treatment efficacy will lead to the most significant gains if they increase the proportion of people diagnosed in BCLC stages 0 and A, as well as improved treatment efficacy for these stages, i.e., the detection of small early stage HCC, and curative HCC treatments. After progression to BCLC stage B and later, the current gains in life expectancy from treatment were low enough that even 25% improvements in either treatment efficacy or ultrasound sensitivity had little impact on population-level outcomes. In addition, because of differences in progression rates by BCLC stage, those with HCC were in BCLC stage A for the longest, which also made it easier to target for intervention (Figure 2A). Although tumor doubling rates are reported in the literature [38], we believe this paper is the first time an analysis of HCC progression has been performed in the BCLC stage-specific framework. While we estimated substantial benefits from improvements to treatments and diagnostic sensitivity in BCLC stage 0 relative to BCLC stage A, in practice, identifying people in BCLC stage 0 is uncommon in Australia. We used Korean BCLC stage-specific treatment efficacy for BCLC stage 0 since there are not much Australian data on this stage [17]. In countries where diagnosis and/or treatment is uncommon during BCLC stage 0, BCLC stage A is likely the best target for improvement of diagnostic and treatment interventions.

Calibrating the model to the data for ultrasounds resulted in comparable ultrasound sensitivities for BCLC stages 0 and A HCC to those reported in the literature. Our model calibration found ultrasound sensitivities of 20.9% (8.0%, 53.7%) and 53.3% (36.1%, 82.0%) for BCLC stages 0 and A, respectively. Most literature places early stage HCC (usually defined by the Milan criteria, but roughly equivalent to BCLC stages 0 and A) ultrasound sensitivities at around 45–65%, with varying results regarding the utility of conducting alpha-fetoprotein blood tests concurrently [24,27]. Other diagnostic tools and biomarkers showing promising results for early stage cancer detection, such as the GALAD (gender, age, L-AFP3, AFP, and des-carboxyprothrombin) risk score, have potential for significant impact through improving surveillance sensitivity [39]. Very few studies report ultrasound sensitivities for HCC that is not early stage, as often it is considered non-curative.

### 4.1. Implications for Future Work

This model can inform future cost-effectiveness analyses. Currently, there is substantial interest in improving ultrasound sensitivity, but our work has raised questions as to whether it may be just as impactful or is more impactful to focus on developing interventions that improve adherence. Further work is warranted to assess the cost-effectiveness of interventions aimed at improving adherence to ultrasound and interventions to improve ultrasound sensitivity. In addition, making the model stochastic would allow for prediction intervals on the various interventions. Although the risk of developing HCC after cirrhosis from hepatitis C after the sustained virologic response is well reported in the literature, this model has been developed to be robust enough to deal with risk profiles that can apply to other diseases with increased incidence of HCC including hepatitis B, metabolic dysfunction-associated steatotic liver disease (MASLD), and alcohol-associated liver disease (ALD).

### 4.2. Limitations

We assumed that all modelled patients received the recommended treatment for their BCLC stage disease and calculated median survival based on that assumption. In reality, people may be downstaged, receive different combinations of sequential treatments, and develop recurrences and new HCC, all potentially contributing to variations in survival over time. Second, BCLC stage-specific treatment efficacy estimates are from a Korean study [17] and may not be generalizable to an Australian population. Third, this is a deterministic model where the whole modelled population was assigned the same probabilities of disease progression and ultrasound adherence. In reality, there may be groups at higher risk of disease progression with different levels of adherence to treatment (for example, those consuming substantial amounts of alcohol). Fourth, interventions in this model only consider years of life gained, not the quality of these years, or the improvements possible through increasing the quality of life for those in the later stages of HCC. Fifth, this analysis was limited in its ability to control for loss to follow-up on ultrasound data, which may have led to some under-reporting of ultrasound adherence. Finally, this analysis has not explored the cost-effectiveness for life years gained specifically, but given the relatively few people with cirrhosis post hepatitis C cure, the fairly targeted surveillance adherence would likely result in a slight increase in additional ultrasounds, which are already recommended as best practice anyway.

## 5. Conclusions

Despite the significant improvements in life expectancy from direct-acting antivirals, regular ultrasound surveillance for those cured of hepatitis C with cirrhosis is essential. Current ultrasound surveillance uptake is suboptimal, and interventions are urgently needed to improve HCC mortality. We show that even modest improvements in adherence to ultrasound appointments can substantially improve life expectancy at a population level.

## Figures and Tables

**Figure 1 cancers-16-02745-f001:**
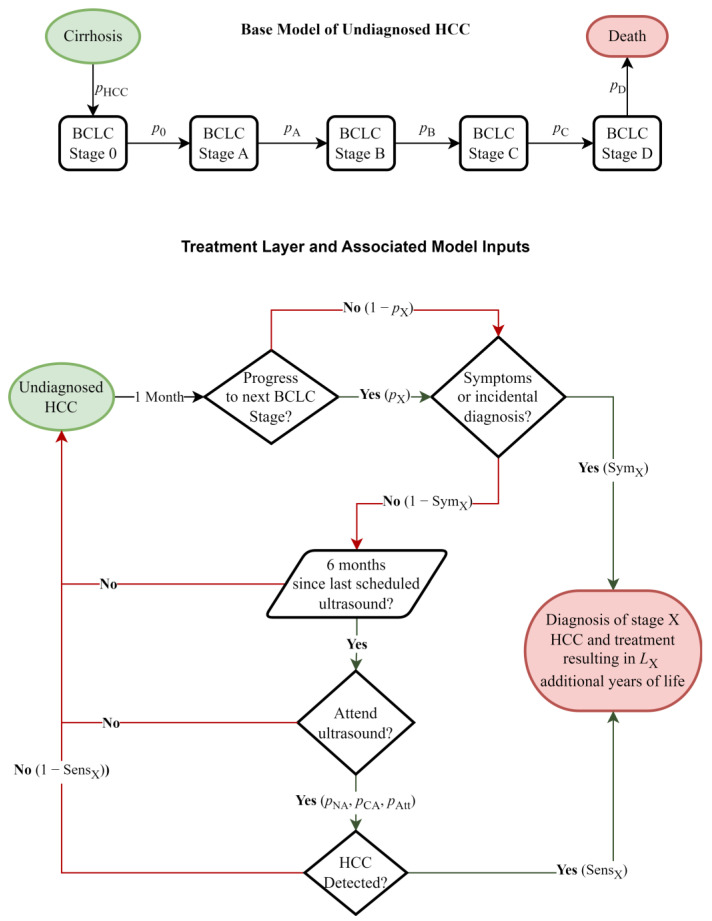
A visual representation of the model. The base model simulates the progression of hepatocellular carcinoma in someone who is undiagnosed (with all people starting in the cirrhosis compartment). The treatment layer describes how once someone develops hepatocellular carcinoma, the model simulates patient diagnosis/treatment. Variables names are defined in Table 2 and Table 3.

**Figure 2 cancers-16-02745-f002:**
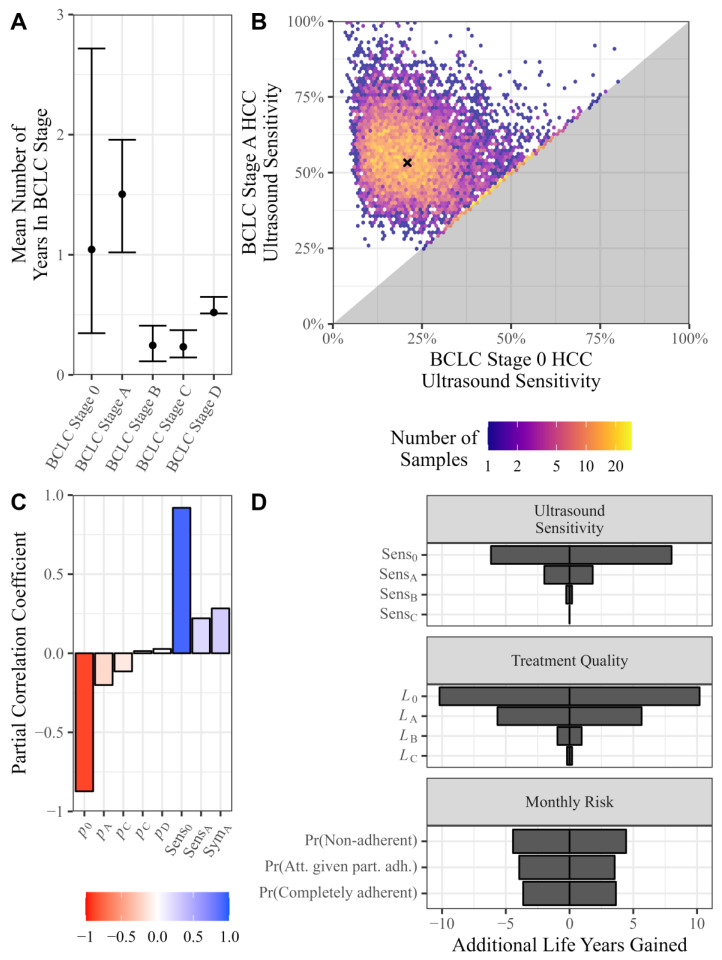
Variables names are defined in Table 2 and Table 3. (**A**) A 95% confidence interval of the mean years spent in each BCLC stage. The dots correspond to the point estimates used in the model. (**B**) Bootstrap estimates for ultrasound sensitivity, with the cross representing the point estimate. The grey area is empty since the calibration assumes BCLC stage 0 ultrasound sensitivity must be lower than BCLC stage A ultrasound sensitivity. (**C**) The partial correlation coefficient between the bootstrap estimates for the model parameters and the status quo outcomes. The bootstrap parameter estimates were logit transformed. (**D**) The effect of a ±25% univariate change in model parameters that could be affected by the intervention and measured relative to baseline in a ten-year simulation of 100 people with cirrhosis.

**Figure 3 cancers-16-02745-f003:**
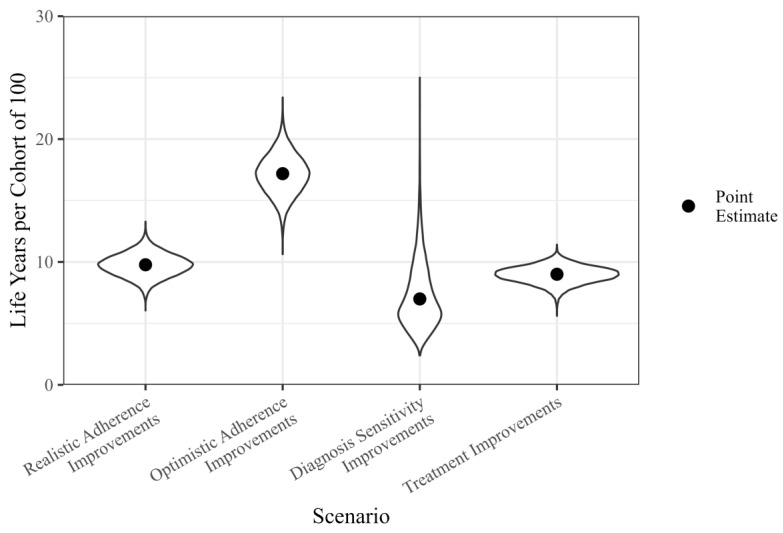
Over a 10-year simulated intervention with 100 people, this figure shows how four scenarios affect the mean additional years of life attributable to each diagnosed case.

**Table 1 cancers-16-02745-t001:** The Barcelona Clinic Liver Cancer (BCLC) staging system. Stages depend on the extent of disease, liver function measured by the Child–Pugh score, and performance status measured by the Eastern Cooperative Oncology Group (ECOG) score. Adapted from Llovet et al. [1].

BCLC Stage	Description
0 (Very early stage)	Single nodule under 2 cm Child–Pugh A, ECOG 0
A (Early stage)	Single nodule, or 2/3 nodules under 3 cm Child–Pugh A–B, ECOG 0
B (Intermediate stage)	Multinodular Child–Pugh A–B, ECOG 0
C (Advanced stage)	Portal invasion, N1, M1 Child–Pugh A–B, ECOG 1–2
D (Terminal stage)	Child–Pugh C, ECOG greater than 2

**Table 2 cancers-16-02745-t002:** Description of non-calibrated model parameter values. Data described as “Informed by” are reasonable estimates given the reported literature. Life expectancy data were derived directly from the data reported by Lee et al. [17].

Parameter	Description	Value	References
$p_{HCC}$	Annual Probability of Developing HCC	2.2%	[18,19,20,21,22]
${Sens}_{B}$	Ultrasound Sensitivity for BCLC B HCC	84.0%	Informed by [8,23,24]
${Sens}_{C}$	Ultrasound Sensitivity for BCLC C HCC	90.0%	Informed by [8,23,24]
${Sens}_{D}$	Ultrasound Sensitivity for BCLC D HCC	95.0%	Informed by [8,23,24]
$L_{0}$	Additional Life Expectancy after Stage 0 Diagnosis in Years	12.5	Derived from [17]
$L_{A}$	Additional Life Expectancy after Stage A Diagnosis in Years	3.6	Derived from [17]
$L_{B}$	Additional Life Expectancy after Stage B Diagnosis in Years	1.7	Derived from [17]
$L_{C}$	Additional Life Expectancy after Stage C Diagnosis in Years	0.25	Derived from [17]
$L_{D}$	Additional Life Expectancy after Stage D Diagnosis in Years	0	Derived from [17]
${Sym}_{0}$	Monthly Probability of Developing Symptoms at Stage 0	0.0%	Informed by [8]
${Sym}_{C}$	Monthly Probability of Developing Symptoms at Stage C	31.9%	Informed by [8]
${Sym}_{D}$	Monthly Probability of Developing Symptoms at Stage D	100.0%	Informed by [8]

**Table 3 cancers-16-02745-t003:** Description of calibrated and calculated model parameters. $p_{0}$, $p_{A}$, $p_{B}$, $p_{C}$, $p_{D}$, ${Sens}_{0}$, ${Sens}_{A}$, ${Sym}_{A}$, and ${Sym}_{B}$ calibrated using data reported by Giannini et al., Khalili et al., and Hong et al. [8,23,25]. $p_{NA}$, $p_{CA}$, and $p_{Att}$ calculated using unpublished data from a tertiary hospital (described in the Appendix A).

Parameter	Description	Point Estimate (95% Confidence Interval)
$p_{0}$	Monthly Probability of Progressing from BCLC 0 to A	8.0% (3.1%, 24.0%)
$p_{A}$	Monthly Probability of Progressing from BCLC A to B	5.5% (4.3%, 8.2%)
$p_{B}$	Monthly Probability of Progressing from BCLC B to C	34.4% (20.3%, 73.7%)
$p_{C}$	Monthly Probability of Progressing from BCLC C to D	35.2% (22.4%, 57.4%)
$p_{D}$	Monthly Probability of Progressing from BCLC D to Death	16.0% (12.8%, 16.3%)
${Sens}_{0}$	Ultrasound Sensitivity for BCLC 0 HCC	20.9% (8.0%, 53.7%)
${Sens}_{A}$	Ultrasound Sensitivity for BCLC A HCC	53.3% (36.1%, 82.0%)
${Sym}_{A}$	Monthly Probability of Developing Symptoms at Stage A	1.0% (0.7%, 1.6%)
${Sym}_{B}$	Monthly Probability of Developing Symptoms at Stage B	8.1% (5.4%, 12.5%)
$p_{NA}$	Probability of being completely non-adherent to surveillance	31.5% (22.9%, 41.1%)
$p_{CA}$	Probability of being completely adherent to surveillance	34.3% (25.4%, 44.0%)
$p_{Att}$	Probability of attending ultrasound if partially adherent	44.9% (35.0%, 55.0%)

**Table 4 cancers-16-02745-t004:** The percentage of simulations in which the scenario in each column results in a higher increase in life years compared to the scenario in each corresponding row. Detailed descriptions of the scenarios are provided in the Section 2.

Scenario	Optimistic Adherence Improvements	Realistic Adherence Improvements	Treatment Improvements
Diagnostic sensitivity improvements	98.4%	80.1%	74.7%
Treatment improvements	100.0%	73.6%	
Realistic adherence improvements	100.0%		

## Data Availability

Data is contained within the article.

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
