# Peer review of "Improving Hepatocellular Carcinoma Surveillance Outcomes in Patients with Cirrhosis after Hepatitis C Cure: A Modelling Study"

_cancers, 2024, doi:10.3390/cancers16152745_

Round 1
Reviewer 1 Report
Comments and Suggestions for Authors
Well designed and written. The importance of the study limitations is clear. The figures and tables are helpful to conceptualize.
Author Response
Thank you for your kind comments.
Reviewer 2 Report
Comments and Suggestions for Authors
Dear authors,
Thank you very much for reviewing your manuscript. I am providing the following comment to address your manuscript and enhance our understanding of your research.
Major Questions:
1. What role does monitoring for hepatocellular carcinoma (HCC) play in cirrhosis patients following a hepatitis C cure?
2. How was the effect of different therapies on HCC outcomes in patients with cirrhosis associated to HCV modelled in this study?
3. What were the main conclusions about how the research cohort's life expectancy was affected by better surveillance adherence?
4. In terms of life years gained, how do gains in adherence compare to increases in ultrasonography sensitivity and treatment efficacy?
5. What findings can be made about the significance of routine HCC ultrasound surveillance for individuals who have cirrhosis and cured hepatitis C?
Minor Questions:
1. What are the current obstacles to patients with HCV-related cirrhosis attaining optimum adherence to biennial HCC screening?
2. What presumptions did the study's deterministic compartmental model make?
3. How was the efficacy of the four distinct interventions—treatment efficacy, diagnostic sensitivity, optimistic adherence, and realistic adherence—quantitatively assessed?
4. What effects will this study have on the way clinicians treat patients with cirrhosis after being cured of HCV?
5. What possibilities for further investigation does this work recommend to enhance the results of HCC surveillance?
Best Regards
Reviewer 3 Report
Comments and Suggestions for Authors
The manuscript by Cumming and coauthors reports an optimized protocol for the surveillance of patients with liver cirrhosis after clearance of hepatitis C virus infection. It is well acknowledged that total clearance of the infection does not always reduce risks of developing hepatocellular carcinoma, especially in patients with advanced fibrosis or cirrhosis. Current strategy involves regular ultrasound monitoring of liver, albeit the period of investigation still remains not quite clear. Moreover, ultrasound investigation remains to be rather low sensitive due to other liver pathologies that alter liver morphology and stiffness. In the presented manuscript the authors provide evidence that such patients should be more adherent to the analysis, and this would provide higher life expectancy for patients that increasing ultrasound sensitivity or the development of more effective treatment regimens. They also provide figures of increased life expectancy for such patients derived from mathematical models based on data on HCC progression from other groups.
The manuscript is clearly written. Actually, this is the rare manuscript that can be published in the current form. I do hope that it will trigger advancement of more effective monitoring of CHC-related cirrhosis patients.
Author Response
Thank you very much for your kind comments.